# Parameter Setting for a Genetic Algorithm Layout Planner as a Toll of Sustainable Manufacturing

**Martin Krajčovič [1], Viktor Hančinský [2], Ľuboslav Dulina [1] , Patrik Grznár [1,\*] , Martin Gašo [1] and Juraj Vaculík [3]**

[1] Faculty of Mechanical Engineering, University of Žilina, Univerzitná 8215/1, 010 26 Žilina, Slovakia; martin.krajcovic@fstroj.uniza.sk (M.K.); luboslav.dulina@fstroj.uniza.sk (Ľ.D.); martin.gaso@fstroj.uniza.sk (M.G.)

[2] GE Aviation s.r.o., Beranových 65, 199 02 Prague 9, Letňany, Czech Republic; kpi@fstroj.uniza.sk

[3] Faculty of Operation and Economics of Transport and Communications, University of Žilina, Univerzitná 8215/1, 010 26 Žilina, Slovakia; juraj.vaculik@fpedas.uniza.sk

\* Correspondence: patrik.grznar@fstroj.uniza.sk; Tel.: +421-41-513-2733

**Abstract:** The long-term sustainability of the enterprise requires constant attention to the continuous improvement of business processes and systems so that the enterprise is still competitive in a dynamic and turbulent market environment. Improvement of processes must lead to the ability of the enterprise to increase production performance, the quality of provided services on a constantly increasing level of productivity and decreasing level of cost. One of the most important potentials for sustainability competitiveness of an enterprise is the continuous restructuring of production and logistics systems to continuously optimize material flows in the enterprise in terms of the changing requirements of customers and the behavior of enterprise system surroundings. Increasing pressure has been applied to projecting manufacturing and logistics systems due to labor intensity, time consumption, and costs for the whole technological projecting process. Moreover, it is also due to quality growth, complexity, and information ability of outputs generated from this process. One option is the use of evolution algorithms for space solution optimization for manufacturing and logistics systems. This method has higher quality results compared to classical heuristic methods. The advantage is the ability to leave specific local extremes. Classical heuristics are unable to do so. Genetic algorithms belong to this group. This article presents a unique genetic algorithm layout planner (GALP) that uses a genetic algorithm to optimize the spatial arrangement. In the first part of this article, there is a description of a framework of the current state of layout planning and genetic algorithms used in manufacturing and logistics system design, methods for layout design, and basic characteristics of genetic algorithms. The second part of the article introduces its own GALP algorithm. It is a structure which is integrated into the design process of manufacturing systems. The core of the article are parameters setting and experimental verification of the proposed algorithm. The final part of the article is a discussion about the results of the GALP application.

**Keywords:** sustainability; genetic algorithm; layout planning; model

## 1. Introduction

Today, the sustainable enterprise needs to use approaches and concepts that allow rapid adaptability of business processes and systems for dynamic environs change. An important concept that creates conditions for quick design reconfiguration of processes and systems is the digital factory.

The digital factory is a concept including a network of digital models, methods, and tools, such as simulation and 3D-visualisation, which are integrated through comprehensive data and flexible

modules management. Products, processes, and resources are modeled based on actual data, in a virtual factory. Based on the actual data and models the planned products and production processes can be improved by use of virtual models until the processes are fully developed, extensively tested, and mostly error-free for their use in a real factory [1].

Manufacturing system's design and layout planning and are basic activities in the digital factory. Their main task is to judge relations of each production system element regarding time and spatial requirements as well as work, technological, handling, control, and other activities inevitable for the rational production process and suitable spatial and time structure of production process.

According to [2] the manufacturing system's design in a digital factory needs three groups of input data. The first group of data is data about the products which will be produced in the production system (assortment, production volume, product structure, physical characteristics of products, demand timeframe, etc.). The second is data about processes used in the production of products (technologies used in the production, steps of processes, operating times, etc.) The last group is data about available/needed resources for production (production machines and facilities, transportation, handling and storage devices, handling units, etc.).

When planning and building manufacturing systems, it is possible to use several approaches whose applicability is dependent on a particular case. Basic approaches according to Lee [3] are as follows:

- Knowledge-based approach: In this approach, the production system is made through gained knowledge, instinct, and common sense. Such systems benefits from the rich knowledge of all current and past employees, but it also has its downside. There is a tendency to use out dated information and overlook new technology and organizational structures. In addition, this approach could be highly unorganized as various workers could have opposing experiences. When planning complex systems, it is advisable to collect knowledge from different perspectives on issues and use them after close examination.
- Cloning: This approach duplicates the existing production system, or which are a part of it. If the existing performance production system is satisfactory and conditions for the planning system are equal, it will be possible to build this production system in a relatively short time, which is the main advantage. However, for the majority of cloning there is limited contribution because the place, process, and people or legislation within each production system could be very different.
- Bottom-Up: The bottom-up and top-down approach starts with details and then it moves step by step up to the level of a whole production plant. The approach is suitable under known conditions and how they should be integrated into a larger group without any change. These conditions are mostly fulfilled in smaller companies in a stable environment. However, the disadvantage of this approach is a long solution period, and until we get to a final layout and plant construction, all details must be integrated. In more complex projects overload of details might occur which leads to worse project clarity.
- Systematic Approach (SLP—Systematic Layout Planning): This approach uses procedures, conventions, and phases which helps the planner to know exactly what to do in each project step. This approach introduces system and structure in planning and also contribution, such as time and work reduction. Primarily, the layout of each block in space is solved.
- Strategic Approach (Top-Down approach): This approach puts emphasis on aims and sets technology and organization in a way they would support each other. This approach starts with a company strategy, plant location selection, and moves forward towards the detailed layout of each element. This approach is direct and with clear aim enabling each project assistant to proceed with same–mutual direction.
- Dominant approach: This approach focuses on a certain form of presentation or company advertisement via a planned plant. It uses an interior or exterior to portray technological innovation, artistic feeling or financial company support.

The layout planning (as a sub-activity of manufacturing system design) is a complex activity involving the optimization of the positions of machines, transportation systems, and workstations [4].

Until recently, classical heuristic approaches (e.g., CORELAP, ALDEP, PLANET, CRAFT, BLOCPLAN) were preferred in optimizing spatial arrangement. Nowadays, layout optimization has been made more efficient by using information technology tools and advanced optimization methods. These methods called metaheuristics have higher quality results compared to classical heuristic methods. Their advantage is the ability to leave specific local extremes and to find a better solution than classical heuristics. One group of metaheuristics are genetic algorithms [5].

Genetic algorithms (GA) are a useful tool for the solution for different tasks in practice. First and foremost, genetic algorithms are used in experiments and simple problems. After verifying the usability of genetic algorithms and the increase of computer technological performance, genetic algorithms started to be used in more complicated tasks. One of the areas of GA practical applications is the design of manufacturing and logistics systems.

One of the first application areas of genetic algorithms in production systems design was production and assembly line balancing and design. In Rekiek [6], the problem of design and optimization of the assembly line was analyzed in detail. However, newly presented multiple objective grouping algorithm (MO-GGA, Multiple Objective Grouping GA) is based on grouping genetic algorithms (GGA) and on the method multiple objective decision PROMETHEE II (Preference Ranking Organization METHod for Enrichment Evaluation). The main difference between grouping and classical genetic algorithms is in gene structure and approach of operators (crossover, mutation, and inversion) of these genes. In GGA, there is a group of objects encrypted in a gene. Apart from an ordinary algorithm, there is the object itself which is encrypted. Thus, there is an optimization of OptiLine software presented in a publication which uses a genetic algorithm.

Hnat [7] underlined a question of assembly line balance through the help of genetic algorithms. In addition, he emphasized a decoder in suitable and proposed technology. Furthermore, use of this application illustrated that a chromosome will not contain an encoded solution, but the information gathered could be proposed as a solution. Moreover, by selecting a suitable representation, it is possible to create a new space instead of the original one.

Kothari [8] designed a genetic algorithm GENALGO for machine arrangement of a given length to line. This algorithm periodically performs a local search for individuals and their suitability increase. Specific function is given as a product sum of all distances between machines and their intensity.

Genetic algorithms for optimizing manufacturing facility layout work [5] summarizes adapted scientific works dealing with various problem solutions of layout optimization. This type of algorithm is used as a solution.

- Suresh with his colleagues used the genetic approach to problem solving of a structural device arrangement with the aim to keep costs for interaction between individual departments to a minimum. Device arrangement problem focuses on finding the best cell arrangement and not solely relying on machines and devices.
- Gupta with his colleagues used a genetic algorithm to distribute products to families and design arrangement between cells. The developed algorithm is focused on cell system arrangement or area arrangement of a production hall rather than the arrangement inside the cells. Neither personal arrangement of machines in cells nor relations between them were considered.
- Two-step hierarchical decomposition approach has been developed, too. First, it is the decision of each object arrangement. For this so-called task, the greedy genetic algorithm is used. Second, it is finding the best disposition arrangement. A genetic algorithm has been used for space solution search. Authors state that for less complex problems, the algorithm offers optimal results and for more serious problems, it overcomes existing methods in speed and quality of the solution.

Kia [9] introduced the model of genetic algorithm use for multi-story objects. In this model, manufacturing cells for pre-defined slots are allocated, but transport between each level has to be

considered—for example, with the help of elevators fixed in the layout. This particular solution does not take into account real machine dimensions in the proposal, nor does it consider relationships between each workplace. However, it is possible to define cell capacity—meaning how many machines a cell can contain. The algorithm, in its fitness function, evaluates transport performance in a cell, between cells and floors. Thus, the algorithm can work in more time periods when it evaluates the price for additional purchase/non-utilization of purchased machines.

Apart from the creation of disposition itself, a genetic algorithm was created for manufacturing cell creation, too. Wu [10] described an algorithm using a two-layered hierarchical scheme, to encode information about machines, products, and also information for dispositional arrangement creation.

Heglas in his work [11] describes the use of discrete simulation together with the use of the evolution method for optimization of the manufacturing system. The output of the work is a designed and verified simulation and an optimization system concept together with a genetic algorithm which is presented by an application form Gasfos2. The application is programmed in the VisualBasic 6.0 language. Thus, it uses a core created by a Galib library (freely accessible library, used for academic purposes, programmed in C++ language which supports working with genetic algorithms) and works with simulation software Witness for optimization manufacturing systems.

The described solutions are primarily focused on the individual cell or department arrangement. They do not take into account real restrictions, showing the inner object arrangement and its entry–exit places. During the search there was no complex system for appropriate integration of such solutions to design process found.

The reason for choice genetic algorithms for the planning of the production layout disposition is dependent on characteristic preferences. It is an especially attribute of genetic algorithms that they leave specific local extremes and find a better solution than classical heuristics.

The author's workplace has long focused on one area of research as well as on the use of genetic algorithms in various areas of industrial engineering. The mentioned workplace has experience with the application of genetic algorithms in the, for example, balancing production lines, parametric simulations, etc. Thus, production layout optimization is another area of research on the application of genetic algorithms.

## 2. Materials and Methods

### 2.1. Methods for Layout Design

The process of layout design requires data from construction and technological preparation of production. Data for the manufacturing and logistics systems design can be divided into two basic groups [12]:

- Numerical data—is mainly used to describe conditions in which the system will operate. They are the basic input for output analyses of the manufacturing and logistics system in compliance with a digital factory concept and the numerical data are structured in three key areas [2]: information about products, which will be made and transported in the manufacturing system (product types, piece lists, construction parameters, planned production volumes, etc.); information about processes of their production (operations, manufacturing and assembling processes, used technologies, time norms, etc.); information about resources for product manufacture (manufacturing machines and equipment, tools, workers, transport, and handling machines, handling units, storage premises, etc.).
- Graphical data—represents a visual display of individual elements of the manufacturing and logistics system which are used mainly in layout design, modeling and simulation of the resultant system.

When we know the need for individual resources of the designed system, material flows and other connections among individual elements, we can begin to design an ideal spatial arrangement of the manufacturing or logistics system.

When proposing an ideal arrangement it is advantageous to use optimization methods and algorithms, which can be classified into [13] four groups:

- Graphical methods—are suitable for the solution of simple problems because when looking for an optimal solution, a graphical presentation of spatial arrangement is used. The following methods belong to this group: The Sankey chart, spaghetti diagram, and relationship diagram, etc.

- Analytical methods—are represented by optimization methods of operational analysis. They are characterized by a mathematical model that describes an objective function and boundary conditions of the problem solution. Their disadvantages are a high demand for calculation, complicated and often impossible mathematical description of real conditions in the system, and low interactivity of a designer with a proposed solution. This group consists of methods of linear and non-linear programming, transport problem, and methods of dynamic programming, etc.

- Heuristic methods—are based on simple algorithms for solution and investigation into the fulfillment of criteria (conditions) given by a particular algorithm. They feature relative simplicity, low demand for computing and high interactivity with a designer (the designer can engage with the solution in any phase). However, there is no guarantee that they will find the global optimum and they are usually unable to determine how close the found solution is to the optimum. Heuristic methods for the proposal of spatial arrangement are divided into construction, change, and combined procedures. Construction procedures are based on gradual insertion of system elements to the layout (it begins with the elements with the highest intensity of transport or with the strongest couplings). The following methods belong to this group: CORELAP, ALDEP, PLANET, MAT, MIP, INLAYT, FLAT, etc. Change procedures go out from the original placement and try to improve through the object interchange. Some examples of the methods are as follows: CRAFT, MCRAFT, MULTIPLE, H63, FRAD, COFAD, etc. Combined procedures use a combination of two approaches mentioned above (it is usually a construction procedure proposing the initial placement and a change procedure for its improvement). Examples of methods: BLOCPLAN, LOGIC, etc.

- Metaheuristic methods—These methods produce results of a much higher quality than classical heuristics. Their advantage is the ability to leave—under certain conditions—found local extrema, which classical heuristics cannot do. The following methods belong to this group: genetic algorithms, simulated annealing, tabu search, Ant Colony optimization, etc.

*2.2. Genetic Algorithms*

The genetic algorithm (GA) belongs to one of the basic stochastic optimization algorithms with distinctive evolutionary features. Nowadays, it is the most used evolutionary optimization algorithm with a wide range of theoretical and practical applications [14,15].

The general procedure of genetic algorithm Figure 1:

1. Initialization—a creation of the initial (zero) population, that usually consists of randomly generated individuals.
2. Start of a cycle—thanks to a certain selection method, a few individuals with a high fitness function are selected from a zero population
3. New individuals—they are generated from selected individuals via the use of basic methods (crossover, mutation, and reproduction), and a new generation is created.
4. Competence calculation of new individuals (fitness function calculation)
5. End of a cycle—a decision-making unit:

    ○ As long as the finishing criterion is not completed, move on to the point no. 2
    ○ If the finishing criterion is finished, the algorithm is completed.

6. End of an algorithm—the individual with the highest competence represents the main algorithm output and the best possible solution found.

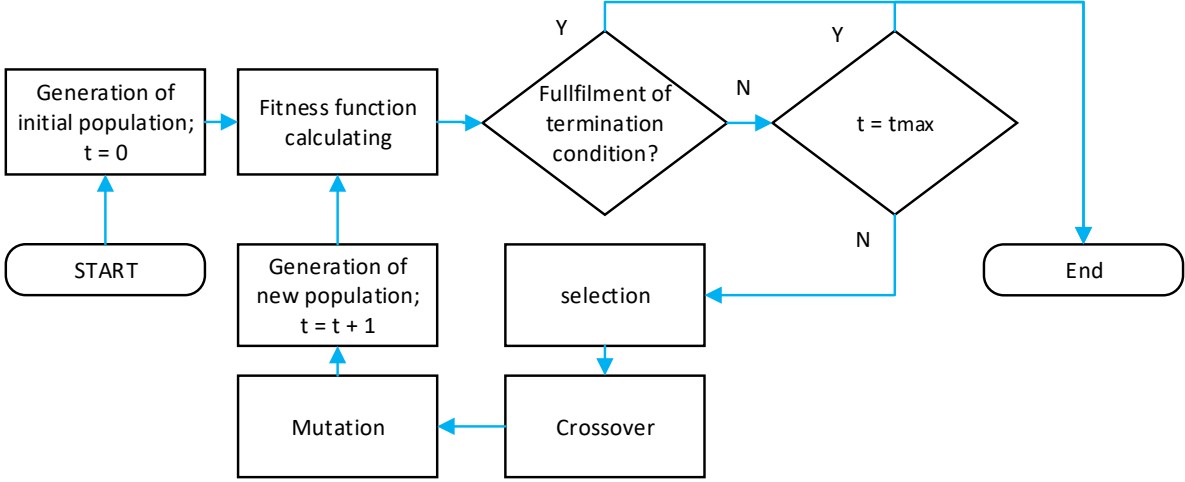

**Figure 1.** Genetic algorithm.

Selecting the appropriate presentation of a problem is the most important part of an application of a genetic algorithm. In a the case of a genetic algorithm, the space of real problem is transformed into the space of strings. These could be, for example, bit-strings, which were one of the first used representations. Real-valued vector representation is the most commonly used for practical issues. In the case of discrete spaces, integer vector could be selected.

Constant population size is regulated in two ways. It is a so-called generation model that replaces the whole population by offsprings (via mutation or recombination). The second option is to keep one part of the previous generation. This is done by elite selection or in other words by individuals with the highest fitness function. By selecting this way, it is guaranteed that the competence of the best individual will continue to improve [16].

Basic parameters of genetic algorithms are [17]:

1. Selection: The selection process of parents for the creation of a next offspring generation. Finding the correct selection pressure is one of the key aspects when looking for an effective solution. High selection pressure leads to a quicker convergence. However, there is a possibility of the algorithm getting stuck in the local extreme. On the other hand, low pressure prolongs the solution time. Multiple criteria are used in selection:

- Fitness—proportionate selection.
- Stochastic universal sampling.
- Rank selection.
- Elitism.
- Steady-state selection.
- Tournament selection.

2. Offspring creation: Two basic operators are used when creating GA offsprings (Figure 2):

- Crossover: A genetic operator which mutually changes chromosome parts.
- Mutation: A genetic operator, used to keep genetic population diversity; mutation will change one or more in chromosomes, which will prevent early convergence of a solution and will provide possibilities of a random search in a closed area of converged population.

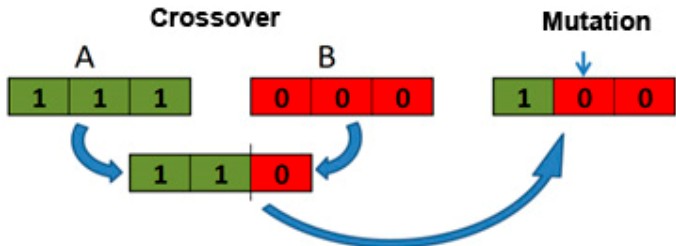

**Figure 2.** Operators: crossover and mutation.

3. Replacement strategies: These decide which offpsrings will have to be replaced in a new generation. One of them uses so-called non-overlapping populations. This means each generation is made of new individuals who are offerings of individuals from a previous generation. However, when the strategy of overlapping populations is used, it is necessary to decide which individuals will be excluded and replaced by new ones.

4. Solution evaluation: Function formulation which contains information about the individual's skillfulness. This formulation is one of the most important points when applying a genetic algorithm to solve problems. Evaluation of solution quality is usually based on fitness function which always returns the real value for each possible solution. The higher or lower (depending on a problem formulation) the value the better the potential solution.

## 3. Results

### 3.1. Genetic Algorithm Integration into the Design Process of Manufacturing Systems

The design approach of manufacturing disposition with the use of genetic algorithms, proposed by authors from this article, requires the realization of the following basic phases as Figure 3 shows:

1. Preparation phase for the disposition arrangement proposal—preparation of numerical data for analysis and layout optimization, graphical data for 2D and 3D model creation of the manufacturing system.

2. Application phase of a genetic algorithm—algorithm core—optimized block layout is its output.

3. Processing phase of designed disposition arrangement in CAD system—the transformation of a proposed block layout into a detailed 3D model of the manufacturing system.

4. Phase of the proposed solution's static verification—verification of a proposed solution based on calculation and analysis of material flows.

5. Phase of proposed solution's dynamic verification—verification of a proposed solution with the use of software simulation [18].

The next chapter of this article contains a detailed description of Phase 2, based on the basic structure of the used genetic algorithm, experimental selection of basic GA parameters and verification of algorithm functionality and comparison of achieved results with a classical heuristics application.

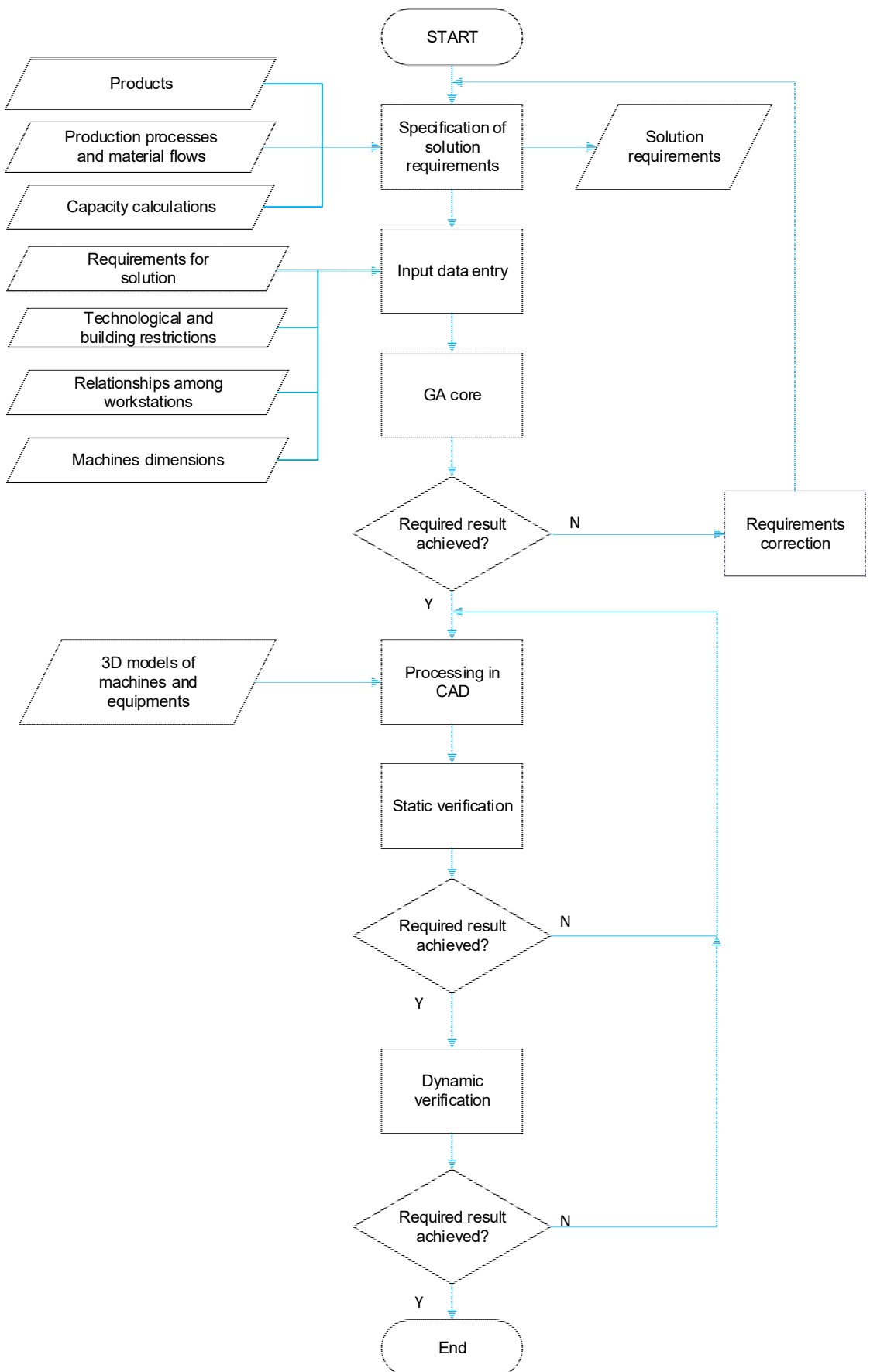

**Figure 3.** Production layout design using a genetic algorithm.

### 3.2. Layout Optimization Using a Genetic Algorithm

The proposed genetic algorithm for layout optimization consists of the following steps as Figure 4 shows: The requirement specification and input value assignment for the GA; core of the GA—optimization of space arrangement; GA procedure completion (finishing requirements).

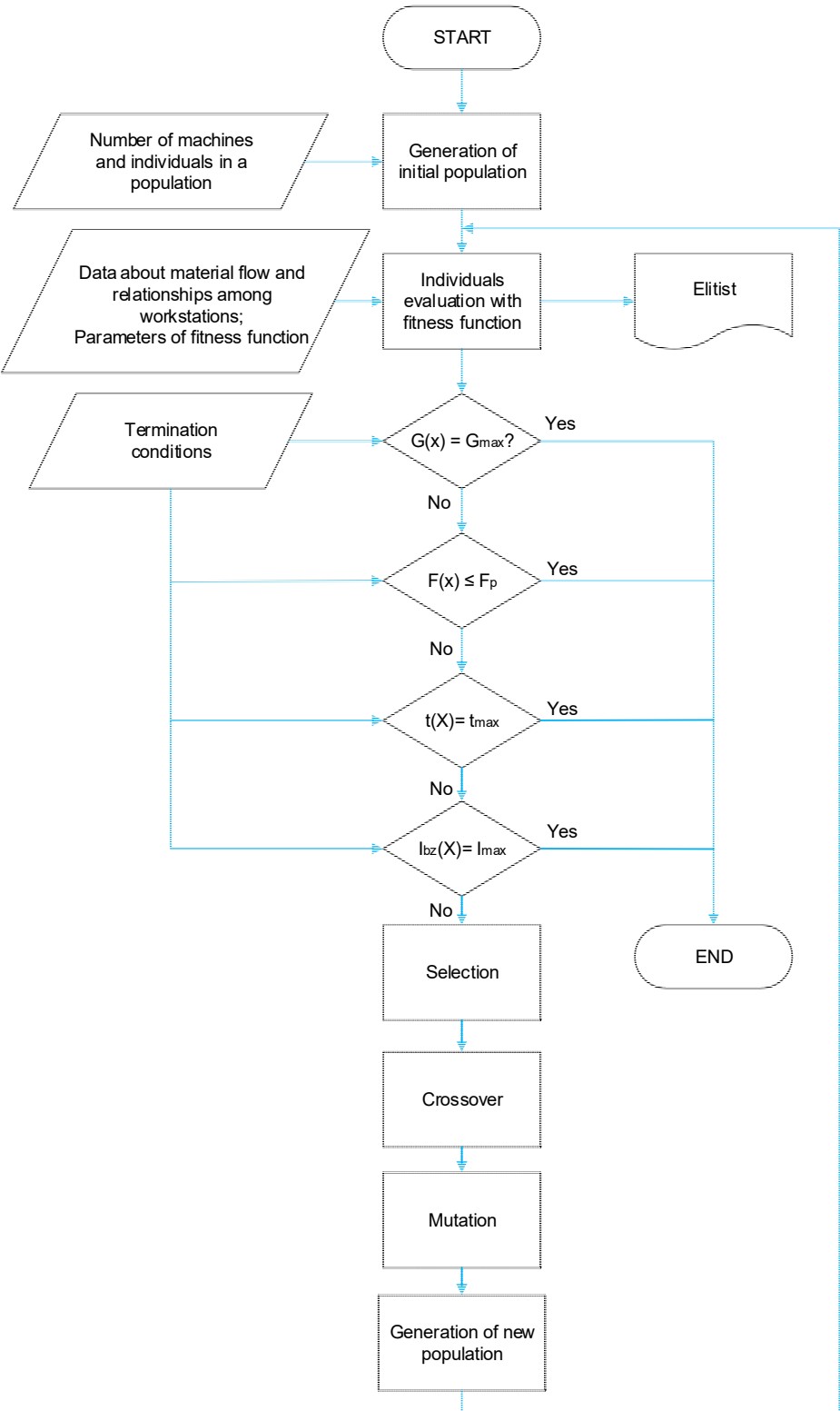

**Figure 4.** Genetic algorithm for layout optimization.

The methodology of layout optimization using the genetic algorithm described in Figure 3 is implemented into a software solution. Software architecture includes six basic software modules that are interconnected. These interconnections were designed within the design of our own planning procedure with the use of a genetic algorithm and provide component activities in the optimization process of the production system and its spatial arrangement Figure 5 [18]. The first software module is a user interface made in a Microsoft (MS) EXCEL environment that allows input parameter setting and summarizes results of optimization. Our own genetic algorithm is programmed in MATLAB. The MATLAB application takes input data from the MS EXCEL table, makes a genetic algorithm procedure and transfers results (block layout, solution progress, and a final value of fitness function) back into the MS EXCEL application. Simultaneously the algorithm generates block layout for an optimal solution into the AutoCAD environment. If the PC workstation has installed the software module from Siemens Tecnomatix (FactoryCAD, FactoryFlow, Plant Simulation) the next phase of manufacturing system design can be realized. It includes the creation of a 3D model of a manufacturing system (FactoryCAD), static analysis of material flow in manufacturing layout (FactoryFlow), and dynamic verification of proposed solution using computer simulation (Plant Simulation).

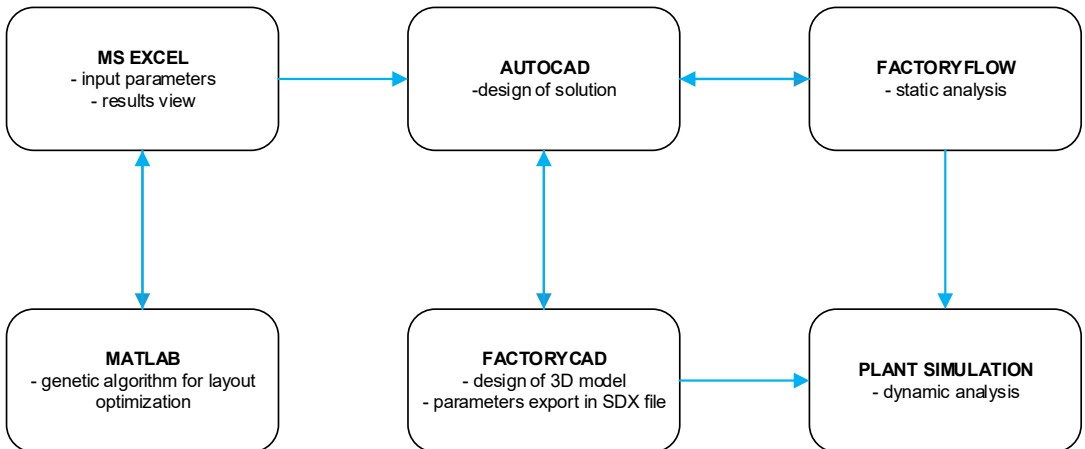

**Figure 5.** Interconnection in a layout design procedure.

3.2.1. Solution Requirement Specification and Input Value Assigning

In the first part of the solution, it is necessary to define basic requirements for the proposed manufacturing disposition. These requirements come from a previous phase of the process and analysis of input data.

For optimization purposes and GA use, it is necessary to set the following parameters [19]:

- Number of placed workplaces, machines, and devices;
- Mutual relations and intensity among workplaces;
- A,E,I,O,U,X coefficients for relation evaluation;
- Ration of fitness function intensity and mutual relations;
- Specification of entry–exit places of a manufacturing system;
- Specification of machines and devices;
- Specification of hall dimensions and potential construction restrictions (walls, columns, corridors).

It is also necessary to set parameters of a genetic algorithm as [20]:

- Maximum number of generations (iterations);
- Number of individuals (solutions) in a generation;
- Selection types, crossover, and mutation of their probability;
- Required value of fitness function (optional information);

- The maximum solution time (optional information);
- The maximum number of generations without solution improvement.

### 3.2.2. Genetic Algorithm for Layout Optimization and Its Basic Parameter Setting

After specification of all input data, our own optimization of space arrangement follows with the help of a genetic algorithm. The basic parts of the GA core as shown in Figure 5 will be explained in the following text.

1. Generation of an initial population

The first step is to create a population that represents a group of solutions which will be further developed. In this solution, an individual is created by genes in the quantity that is equivalent to the value of placed machines. These can have a value of 1 up to n, where n is equal to the number of deployed machines. A sequence of individual genes corresponds with a sequence of where machines will be placed in the proposal. Next, there is one gene in each individual reserved for a pattern definition by which workplaces will be included in the proposal as it shows Figure 6.

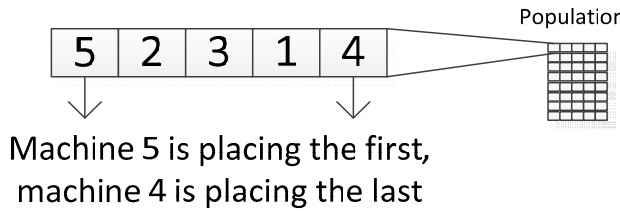

**Figure 6.** Interpretation of machine order and identification.

The total matrix dimension corresponding to a population in one generation [20] is, therefore, the number of individuals in a generation * (number of placed machines + 1).

2. Individual evaluation of fitness function

After having created a population, it is necessary to evaluate a fitness function. The resulting fitness function was designed as a sum of two components with verified weight. Verification was done according to the intensity of material flow and distance ($f_{ID}$) and according to relations and distance ($f_V$).

Equation (1)—Evaluation according to the intensity and distance [19]:

$$f_{ID} = \sum_{i,j=1}^{i,j=n} D_{ij} * I_{ij} \tag{1}$$

where n = number of placed machines; D = right-angle distance between workplaces ($D_{ij} = |x_i - x_j| + |y_i - y_j|$); I = intensity between workplaces i and j.

See note: Right angle distance was chosen for the distance evaluation of a closer real state rather than straightforward distance.

Equations (2) and (3)—Evaluation according to relations and distance [21]:

$$f_V = \sum_{i,j=1}^{i,j=n} V_{ij} * D_{ij} \quad \text{for } V_{ij} \geq 0 \tag{2}$$

$$f_V = \sum_{i,j=1}^{i,j=n} \frac{V_{ij}^2}{D_{ij}} \quad \text{for } V_{ij} < 0 \tag{3}$$

where n = number of placed machines; D = right angle distance between workplaces i–j ($D_{ij} = |x_i - x_j| + |y_i - y_j|$); V = evaluation coefficient of a relation between workplaces i and j (A,E,I,O,U,X).

Equation (4)—Final fitness function value is set as:

$$\min: f = \alpha * f_{ID} + (1 - \alpha) * f_V \tag{4}$$

where $\alpha$ = ratio coefficient of partial fitness functions ($\alpha \in <0;1>$)

Various restrictions are checked in layout construction and the algorithm itself:

- Verification if each object is not mutually overlapping;
- Verification of placed objects in a defined space (dimensions of a production hall);
- Verification of the production hall height restriction;
- Verification of a production hall's height restriction;
- Verification of restrictions regarding transport street arrangement in the production hall;
- Verification of restrictions regarding the definition of the selected object's fixed position in a production hall.

After evaluating all individuals by a fitness function, the best solution is identified and saved in a given generation—an elite individual with his or her reached value and the average value of the fitness population. This data could be displayed during algorithm operation after each generation, to track solution progress. After completion, it is also possible to display a progress graph of average and elite fitness values.

3. Decision-making blocks

In this step, it is necessary to compare specific conditions for algorithm termination in four decision-making blocks. The first condition is to reach the maximum number of generation (iteration) $G_{max}$. The second condition is to reach or exceed the highest permissible fitness value $f_p$. The third condition is to reach maximum solution time $t_{max}$. The last condition is to exceed set iteration number ($I_{max}$) without improving a reached solution.

The last condition was integrated into the proposal to prevent extensive calculation time if the required or unachievable fitness function value $f_p$ is not set and fitness function value is not improving. Therefore, there is an assumption, that the extreme has been found in a group of solutions.

When meeting any out of four stated conditions, the genetic algorithm is completed.

4. Selection

In case none of the finishing criteria was fulfilled, the algorithm continues by selection, in other words, by selecting individuals who will crossbreed and eventually mutate between each other. For such a solution, the roulette rule was selected. Probability selection was proportional to an individual's achieved suitability. This form was chosen based on a better possibility to search a complex set of solutions when later combining parents and their evaluation as well as their calculation speed [3].

To prevent early convergence, suitability of individuals was integrated into the algorithm via the help of sigma scaling. The average expected number of generated offsprings with sigma scaling is $p_{(i,g)}$ from an individual *i* in generation *g* given as Equation (5):

$$p(i,g) = \begin{cases} 1 + \frac{f_{(i,g)} - \overline{f}_{(g)}}{ks * \sigma_{(g)}} & for\ \sigma_{(g)} \neq 1 \\ 1 & for\ \sigma_{(g)} = 0 \end{cases} \tag{5}$$

where $f_{(i,g)}$ = fitness i of an individual in generation g; $f_{(g)}$ = average population fitness in generation g; $k_{\check{s}}$ = coefficient for sigma scaling; $\sigma_{(g)}$ = determinant population deviation in generation g.

For a sigma scaling coefficient $k_s = 1$, an individual rated by the suitability of its standard deviation being closer to a required extreme as the average population suitability will on average produce two offsprings for a new population. The higher the value $k_s$, the lower the selective pressure.

After remapping, it is possible to select a choice of parents (Figure 7) either by the classical roulette mechanism (generation of random numbers) or by stochastic universal sampling (they are generated uniformly spread indicators that will choose parents in one iteration).

After selecting, pairing follows, where Parent 1 and Parent 2 will be randomly selected from chosen individuals. These should make Offspring 1 and 2.

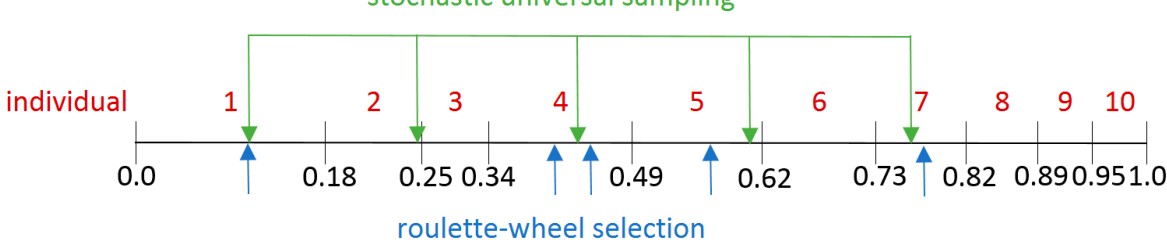

**Figure 7.** Between roulette-wheel selection and stochastic universal sampling.

5. Crossover

To prevent a duplicate of identical machines in crossover or omission of the same machines from the genetic chain, a mechanism of partially matched crossover was designed. This type of crossover has within its procedure implemented measures. This guarantee that each coded solution will have its machine only once [21].

A procedure of partially matched crossover (Figure 8) is as follows:

1. Generation of two random points delimiting genes, parents will mutually exchange.
2. Pairing of gene values that have been exchanged.
3. Adding parent values to genes where conflicts do not occur.
4. Use of paired values for conflict genes.

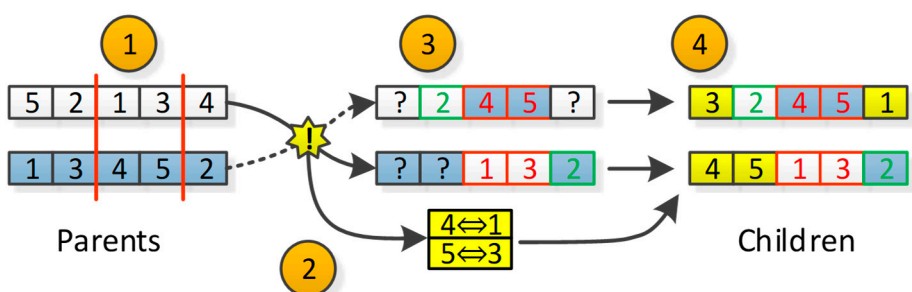

**Figure 8.** Matched crossing.

It is also necessary to set the crossover's probability and to determine an optimal value range of probability. A series of experiments were carried out. There were two case studies, and the only changed parameter was the probability of crossover. [22] Mutation was switched off and the finishing criterion was set to reach 200 generations. The fitness function and generation value were reached through closely observed parameters. In Experiment 1, 20 machines were placed into a layout, and in Experiment 2, 28 machines were placed. Results of experiments can be seen in Tables 1 and 2.

The optimal crossover probability was set in the range from 75% to 95%, based on a series of experiments. The probability of 100% was not taken into consideration because it was possible for the small part of the previous population to survive.

**Table 1.** Various crossover probabilities and their results—Experiment 1.

| | Number of Crossover | 50 | 55 | 60 | 65 | 70 | 75 | 80 | 85 | 90 | 95 | 100 |
|---|---|---|---|---|---|---|---|---|---|---|---|---|
| Fitness | 1. run | 1,786,250 | 1,801,250 | 1,882,500 | 1,801,250 | 1,960,000 | 1,901,250 | 1,767,212 | 1,892,500 | 1,862,500 | 1,850,000 | 1,735,000 |
| | 2. run | 1,815,033 | 1,927,500 | 1,946,250 | 1,843,750 | 1,982,500 | 1,806,250 | 1,771,250 | 1,825,000 | 1,851,250 | 1,755,000 | 1,817,212 |
| | 3. run | 1,842,500 | 1,935,000 | 1,847,500 | 1,841,250 | 1,811,250 | 1,704,712 | 1,777,212 | 1,763,750 | 1,850,000 | 1,825,000 | 1,935,000 |
| | 4. run | 1,865,000 | 1,935,000 | 1,956,250 | 1,875,000 | 1,928,462 | 1,891,250 | 1,835,962 | 1,765,962 | 1,936,250 | 1,925,962 | 1,837,500 |
| | 5. run | 1,877,500 | 1,840,962 | 1,853,750 | 1,860,962 | 1,742,500 | 1,793,250 | 1,825,962 | 1,916,250 | 1,654,712 | 1,825,000 | 1,866,250 |
| | 6. run | 1,741,250 | 1,858,750 | 1,773,750 | 1,891,250 | 1,900,000 | 1,794,712 | 1,782,503 | 1,733,750 | 1,780,000 | 1,898,750 | 1,824,712 |
| | **Average** | **1,821,256** | **1,883,077** | **1,876,667** | **1,852,244** | **1,887,452** | **1,815,237** | **1,793,350** | **1,816,202** | **1,822,452** | **1,846,619** | **1,835,946** |
| Generation | 1. run | 145 | 138 | 109 | 105 | 69 | 70 | 102 | 128 | 71 | 77 | 55 |
| | 2. run | 136 | 121 | 91 | 97 | 56 | 90 | 93 | 79 | 67 | 100 | 132 |
| | 3. run | 128 | 108 | 104 | 96 | 90 | 91 | 69 | 83 | 85 | 75 | 88 |
| | 4. run | 117 | 104 | 105 | 98 | 115 | 53 | 93 | 84 | 136 | 91 | 111 |
| | 5. run | 112 | 133 | 74 | 132 | 106 | 92 | 79 | 79 | 90 | 102 | 73 |
| | 6. run | 94 | 113 | 131 | 121 | 90 | 127 | 125 | 134 | 88 | 58 | 91 |
| | **Average** | **122.0** | **119.5** | **102.3** | **108.2** | **87.7** | **87.2** | **93.5** | **97.8** | **89.5** | **83.8** | **91.7** |

**Table 2.** Various crossover probabilities and their results—Experiment 2.

| | Number of Crossover | 50 | 55 | 60 | 65 | 70 | 75 | 80 | 85 | 90 | 95 | 100 |
|---|---|---|---|---|---|---|---|---|---|---|---|---|
| Fitness | 1. run | 2,120,279 | 1,973,988 | 2,074,141 | 2,065,830 | 2,014,884 | 2,087,429 | 2,197,170 | 2,241,621 | 2,034,974 | 2,027,345 | 2,036,681 |
| | 2. run | 2,067,359 | 1,965,956 | 2,218,102 | 2,196,600 | 2,179,517 | 2,035,914 | 2,135,551 | 1,972,258 | 2,124,068 | 2,085,635 | 2,209,369 |
| | 3. run | 2,140,407 | 2,268,313 | 2,180,144 | 2,090,574 | 2,180,138 | 2,215,854 | 2,169,616 | 1,836,980 | 2,091,044 | 2,117,620 | 2,034,550 |
| | 4. run | 2,258,198 | 2,090,424 | 2,277,779 | 2,194,130 | 2,250,233 | 2,126,470 | 2,180,122 | 2,105,810 | 2,025,580 | 2,099,738 | 2,157,909 |
| | 5. run | 2,208,670 | 2,227,285 | 2,133,529 | 2,024,841 | 2,158,330 | 2,164,587 | 2,197,480 | 2,062,865 | 2,133,644 | 2,085,187 | 2,143,426 |
| | 6. run | 2,219,823 | 2,059,236 | 2,150,834 | 2,188,592 | 2,202,106 | 2,172,660 | 2,135,509 | 1,975,300 | 2,022,912 | 2,165,858 | 2,216,239 |
| | **Average** | **2,169,123** | **2,097,534** | **2,172,422** | **2,126,761** | **2,164,201** | **2,133,819** | **2,169,241** | **2,032,472** | **2,072,037** | **2,096,897** | **2,133,029** |
| Generation | 1. run | 142 | 105 | 84 | 103 | 136 | 122 | 117 | 116 | 82 | 110 | 100 |
| | 2. run | 88 | 113 | 115 | 137 | 151 | 90 | 100 | 121 | 92 | 82 | 56 |
| | 3. run | 169 | 110 | 120 | 131 | 102 | 81 | 75 | 129 | 74 | 103 | 129 |
| | 4. run | 95 | 190 | 68 | 139 | 53 | 105 | 60 | 77 | 144 | 82 | 89 |
| | 5. run | 121 | 99 | 161 | 134 | 137 | 98 | 130 | 87 | 134 | 109 | 61 |
| | 6. run | 134 | 86 | 160 | 88 | 103 | 140 | 92 | 95 | 87 | 59 | 59 |
| | **Average** | **124.8** | **117.2** | **118.0** | **122.0** | **113.7** | **106.0** | **95.7** | **104.2** | **102.2** | **90.8** | **82.3** |

## 6. Mutation

After the crossover, mutation follows. The heuristic insert mutation was adopted to better ensure population diversity and individual qualities after mutation as well as satisfy the process constraints. [18] However, in this type of solution encoding, traditional mutation or in other words value change of a random gene, is out of the question. This would automatically require remedial measures to eliminate duplication or not classified machines. That is why mutation via the help of inversion or exchange was selected in Figure 9. Due to inversion of a rather big intervention into solution, the probability was divided for exchange or inversion in 80:20 ratio.

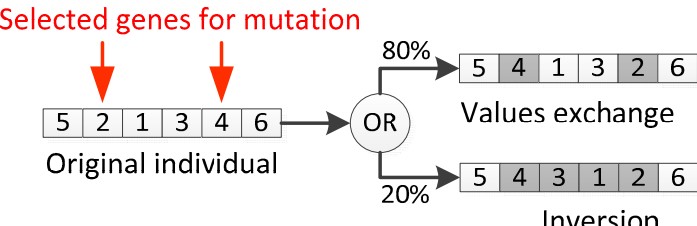

**Figure 9.** Principle in the proposed algorithm.

Furthermore, it was necessary to set the probability in mutation and the majority of resources state range, depending on the type of problem, from 0.1% to 5%. However, in applications for object arrangement, some resources state probability values of up to 20% [23]. Due to this, the optimal probability ratio of mutation for this specific problem was selected similarly as in crossover. The set of experiments for the same initial conditions (Experiment 1—20 machines, Experiment 2—28 machines) as was the case for selecting optimal crossover probability values were carried out. However, the difference was the mutation probability, which was substituted. In the experiments, crossover probability was set to a constant value of 0.85. The results of the experiments are stated in Tables 3 and 4.

The results of experiments state that with low probability mutation, functioning converges later as it is mainly dependent on randomly generated initial population and crossbreeding in all iterations. Only a small number of individuals is modified by mutation operators. With increasing mutation probability, race converges on average earlier with a higher quality solution, although it is accompanied by a higher generation dispersion of a found solution. This is caused by mutation randomness. The optimal value probability range of mutation was set between 0.05 and 0.15. As we want to avoid the algorithm going into a random search, we do not recommend higher probabilities of initial settings.

One of the conditions of algorithm functioning termination is crossing the fixed number of iterations ($I_{max}$) without improving the reached solution. In case the extreme has been achieved, it is advised to verify if it is only a local extreme. To verify this, the variable mutation was implemented in the algorithm. This variable mutation increases the probability of its application with an increasing number of interactions, without any improvement. In the basic setting—when functioning finishes after 100 interactions without any improvement, after 70 iterations ($I_{bz} = 70$), there is a mutation probability increased to 1.5 multiple of the original value. After 80 iterations it is 1.875 multiple of the original value, and eventually, after 90 iterations, it is totally 2.34 multiple of the original value. If there is a different setting for the number of iterations without any improvement, Imax is the variable probability of mutation proportionally recalculated.

## 7. Making of a new generation

Following the genetic operator activity, parents are replaced by offsprings. In case elitism is used in suitability evaluation and the best possible solution has been saved, this individual replaces one of the offsprings with the worst suitability.

After this step, the algorithm goes back to evaluating new individuals through the help of the fitness function. Furthermore, the algorithm keeps repeating in cycles until one of the finishing conditions is fulfilled in Figure 6.

**Table 3.** Results of various mutation probabilities—Experiment 1.

| | Number of Crossover | 0.01 | 0.02 | 0.04 | 0.05 | 0.08 | 0.1 | 0.12 | 0.15 | 0.18 | 0.2 | 0.22 | 0.25 |
|---|---|---|---|---|---|---|---|---|---|---|---|---|---|
| Fitness | 1. run | 1,826,250 | 1,975,000 | 1,801,250 | 1,753,750 | 1,835,000 | 1,778,750 | 1,771,250 | 1,791,250 | 1,800,000 | 1,846,250 | 1,842,500 | 1,753,750 |
| | 2. run | 1,922,500 | 1,795,000 | 1,845,000 | 1,787,500 | 1,833,750 | 1,796,254 | 1,813,750 | 1,915,000 | 1,813,750 | 1,785,000 | 1,768,750 | 1,761,250 |
| | 3. run | 1,841,250 | 1,892,500 | 1,872,500 | 1,770,000 | 1,783,750 | 1,817,500 | 1,836,250 | 1,752,500 | 1,771,250 | 1,820,000 | 1,760,000 | 1,752,500 |
| | 4. run | 1,875,000 | 1,901,250 | 1,790,000 | 1,773,750 | 1,921,250 | 1,820,000 | 1,782,500 | 1,763,750 | 1,812,500 | 1,727,500 | 1,817,500 | 1,796,250 |
| | 5. run | 1,977,500 | 1,793,750 | 1,926,250 | 1,893,750 | 1,896,250 | 1,750,000 | 1,772,500 | 1,717,500 | 1,820,000 | 1,756,250 | 1,762,500 | 1,801,250 |
| | 6. run | 1,893,750 | 1,897,500 | 1,852,500 | 1,855,000 | 1,743,750 | 1,905,000 | 1,797,500 | 1,847,500 | 1,772,500 | 1,682,500 | 1,717,500 | 1,687,500 |
| | Average | **1,889,375** | **1,875,833** | **1,847,917** | **1,805,625** | **1,835,625** | **1,814,251** | **1,795,625** | **1,797,917** | **1,798,333** | **1,769,583** | **1,778,125** | **1,758,750** |
| Generation | 1. run | 166 | 166 | 0:00 | 92 | 141 | 155 | 81 | 63 | 102 | 76 | 83 | 116 |
| | 2. run | 133 | 159 | 69 | 120 | 62 | 111 | 109 | 62 | 119 | 82 | 79 | 76 |
| | 3. run | 117 | 94 | 166 | 89 | 68 | 136 | 175 | 99 | 61 | 65 | 83 | 83 |
| | 4. run | 73 | 81 | 111 | 122 | 119 | 102 | 112 | 78 | 74 | 86 | 66 | 116 |
| | 5. run | 170 | 111 | 138 | 81 | 92 | 130 | 54 | 80 | 65 | 47 | 101 | 57 |
| | 6. run | 157 | 110 | 158 | 101 | 117 | 85 | 70 | 149 | 60 | 62 | 75 | 49 |
| | Average | **136.0** | **120.2** | **125.5** | **100.8** | **99.8** | **119.8** | **100.2** | **88.5** | **80.2** | **69.7** | **81.2** | **82.8** |

**Table 4.** Results of various mutation probabilities—Experiment 2.

| | Number of Crossover | 0.01 | 0.02 | 0.04 | 0.05 | 0.08 | 0.1 | 0.12 | 0.15 | 0.18 | 0.2 | 0.22 | 0.25 |
|---|---|---|---|---|---|---|---|---|---|---|---|---|---|
| Fitness | 1. run | 2,205,047 | 2,094,709 | 2,150,884 | 2,031,828 | 2,024,367 | 2,097,308 | 2,040,704 | 2,023,408 | 1,975,890 | 2,106,751 | 2,167,212 | 2,007,581 |
| | 2. run | 2,293,938 | 2,224,044 | 2,068,203 | 2,032,667 | 2,119,026 | 2,138,655 | 2,057,521 | 2,174,757 | 2,153,584 | 2,081,075 | 2,004,717 | 1,923,141 |
| | 3. run | 2,174,356 | 2,223,835 | 2,236,219 | 2,108,171 | 2,107,460 | 2,183,970 | 1,984,400 | 2,184,988 | 2,230,222 | 2,078,632 | 2,073,966 | 2,030,595 |
| | 4. run | 2,209,866 | 2,101,225 | 2,059,109 | 2,070,532 | 2,038,591 | 2,235,402 | 2,240,006 | 2,177,516 | 2,089,100 | 2,151,902 | 2,261,843 | 2,034,200 |
| | 5. run | 2,036,578 | 2,133,150 | 2,096,508 | 1,983,330 | 2,165,419 | 2,101,584 | 2,059,324 | 2,091,831 | 2,133,639 | 2,147,631 | 1,945,574 | 1,978,386 |
| | 6. run | 2,002,145 | 2,073,131 | 2,286,277 | 2,131,030 | 2,098,622 | 2,165,754 | 2,128,676 | 2,244,776 | 2,095,610 | 2,150,529 | 2,173,074 | 2,001,748 |
| | Average | **2,153,655** | **2,141,682** | **2,149,533** | **2,059,593** | **2,092,248** | **2,153,779** | **2,085,105** | **2,149,546** | **2,113,008** | **2,119,420** | **2,104,398** | **1,995,942** |
| Generation | 1. run | 162 | 92 | 122 | 106 | 144 | 116 | 113 | 81 | 182 | 105 | 81 | 86 |
| | 2. run | 106 | 190 | 94 | 105 | 96 | 161 | 97 | 90 | 99 | 75 | 133 | 77 |
| | 3. run | 97 | 85 | 158 | 73 | 100 | 74 | 167 | 77 | 99 | 43 | 118 | 126 |
| | 4. run | 96 | 83 | 103 | 115 | 81 | 94 | 74 | 61 | 135 | 103 | 95 | 125 |
| | 5. run | 177 | 89 | 181 | 91 | 106 | 98 | 107 | 119 | 147 | 86 | 77 | 71 |
| | 6. run | 97 | 67 | 89 | 176 | 112 | 124 | 94 | 93 | 142 | 127 | 84 | 132 |
| | Average | **122.5** | **101.0** | **124.5** | **111.0** | **106.5** | **111.2** | **108.7** | **86.8** | **134.0** | **89.8** | **98.0** | **102.8** |

## 8. Genetic algorithm finishing

In decision-making blocks, each genetic algorithm cycle checks whether one of the finishing conditions has not been fulfilled: achieving the maximum number of generations (iterations); achieving or exceeding the highest permissible fitness value; achieving the maximum solution time; exceeding the set number of iterations without improvement.

If some of the finishing conditions were fulfilled, the activity of a genetic algorithm will finish. After completion of its activities, there are generated outputs in the user interface (Figure 10): block layout; achieved fitness value and information in which iteration it was achieved; graph showing the progress of average and elite fitness population values.

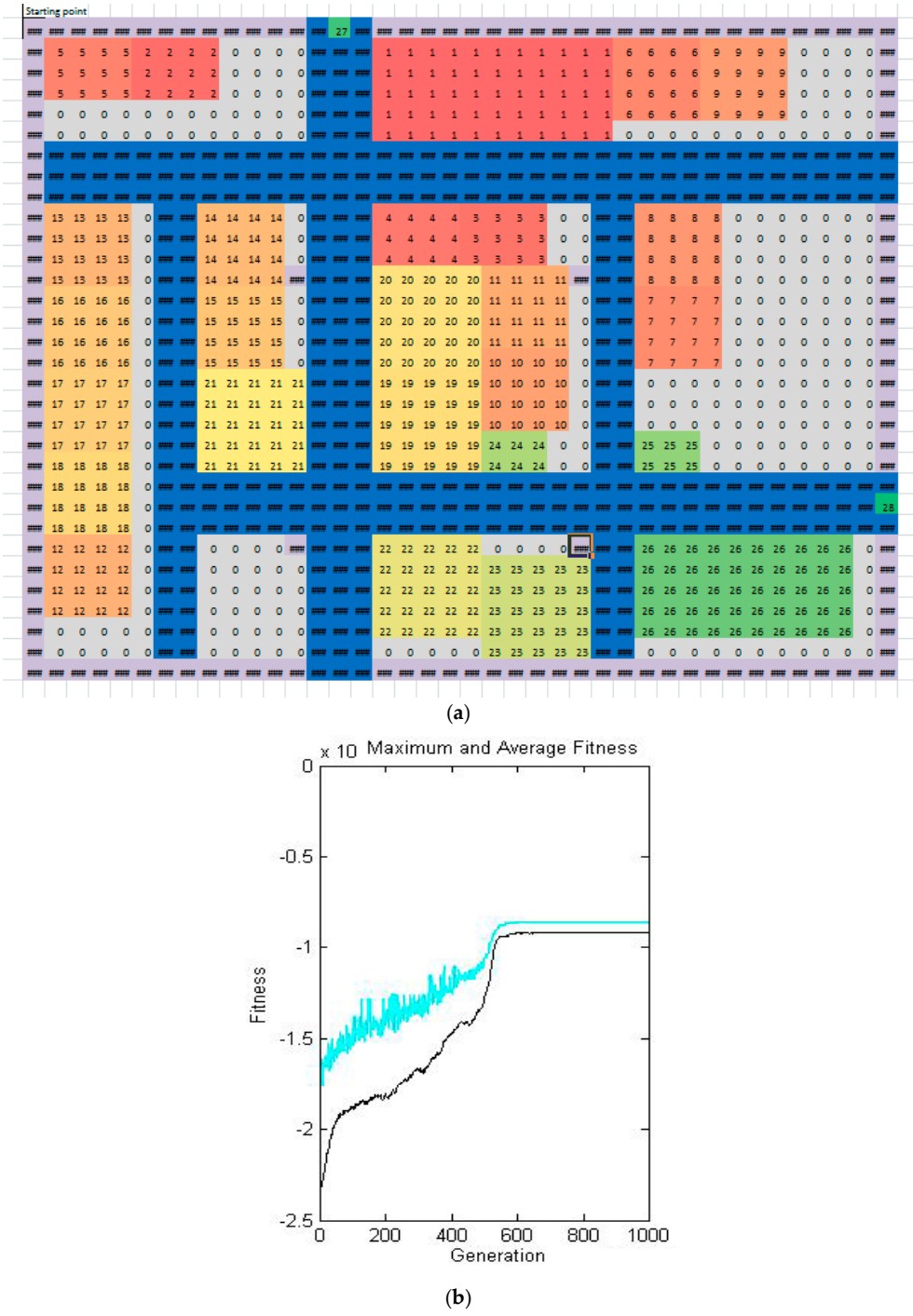

**Figure 10.** Results of genetic algorithm (**a**) block layout; (**b**) fitness value chart.

In the final phase, the user will decide whether the solution proposed by the genetic algorithm fulfilled all its requirements. If not, it is necessary to closely specify requirements and repeat the generation of an optimal layout. If requirements were fulfilled, the methodology continues by the result of processing in a CAD system.

3.2.3. Experimental Verification of Genetic Algorithm and Result Compared with the Use of Classical Heuristic

To check the functionality of the proposed GALP algorithm (Genetic Algorithm Layout Planner) a series of experiments were carried out. These results were then compared with optimization results with the help of heuristic according to Murat (sequence–pair approach). Heuristic according to Murat has been selected because it is believed the heuristic approach is implemented in Factory PLAN/OPT module, which is a part of Factory Design and Optimization package from Siemens Tecnomatix. The final PLAN/OPT and genetic GALP algorithm proposals were subsequently compared in FactoryFLOW software.

A common characteristic for both algorithms is the block layout output. Both algorithms require a finishing requirement and total time of algorithm functioning. For more complex result comparison, experiments were carried out for 1.5, 10, and 20 min.

Our own experiments were carried out for two types of inputs. Our own experiments were carried out for two types of inputs. In Case 1 a simple manufacturing system that represents a small manufacturing system was generated. This system contains three product families (each of the family has minimum of eight process steps) and 24 workplaces. The number of workplaces was determined based on the planned annual production volume of product families. In Case 2 a complex manufacturing system with nine product families and 60 workplaces. This system represents a large manufacturing system in practice. Of course, there are also much larger manufacturing systems in practice, howe;ver, these systems can be segmented into smaller, mutually independent production groups, based on the classification of the manufactured products.

Evaluation of the application results of the classic heuristics and the GALP algorithm was made on the calculation basis of three basic material flow parameters, which are automatically calculated by Tecnomatix FactoryFLOW software. The first parameter is distance. This parameter determines the total amount of distance traveled in the production layout when considering the right-hand distances and the total number of journeys between different twin of workstations. The second parameter is the cost. This parameter determines the total transport costs on the basis of total distance traveled, the type of transport equipment, and the fixed and variable rates of transport costs. The third parameter, time, specifies total shipment time based on the known distance traveled, the type of transport equipment, defined transport cycle structure (i.e., load–drive-unload) and the time parameters of the transport cycle (i.e., load time, unload time, transportation speed).

Case 1 results are shown in Tables 5–7. The graphical expression of comparison results shown (Figure 11). These experimental results indicate that GALP achieved better results in all cases than the PLAN/OPT algorithm, which, due to unknown reasons, did not keep workplace dimensions in some cases (Figure 12). Case 1 results show that the difference between the classic heuristics (PLAN/OPT) and the GALP algorithm also depends on the calculation time and, hence, on the total number of iterations used by the algorithm. For the shortest calculation time of 1 min, the differences between results at 1.21% (distance and cost savings) and 0.63% (time savings) are in favor of the proposed GALP algorithm. At the longest calculation time (20 min), savings increased to 1.82% (distance and cost savings) and 2.32% (time savings). GALP has also proposed solutions preferring singular direction of material flow with minimum crossing or backward material flow. Due to comparing both algorithms, no restrictions have been imposed on workplace arrangement. However, the GALP algorithm enables basic restriction definition in the layout (production hall dimensions, the height of spaces, material component arrangement, fixed installations or transport corridors in the layout).

**Table 5.** Experiment results for GALP.

| Parameter | Time Calculation | Distance (m) | Costs (EUR) | Time (min) |
|---|---|---|---|---|
| | 1 min | 571,360.03 | 25,434.76 | 69,708.00 |
| Achieved results | 5 min | 510,552.24 | 25,023.22 | 67,682.66 |
| | 10 min | 441,472.39 | 24,825.88 | 67,536.68 |
| | 20 min | 430,341.00 | 24,755.86 | 67,214.88 |

**Table 6.** Experiment results for PLAN/OPT.

| Parameter | Time Calculation | Distance (m) | Costs (EUR) | Time (min) |
|---|---|---|---|---|
| | 1 min | 669,925.47 | 25,746.26 | 70,146.72 |
| Achieved results | 5 min | 633,664.26 | 25,668.70 | 70,281.26 |
| | 10 min | 548,718.83 | 25,284.36 | 68,976.65 |
| | 20 min | 529,770.15 | 25,214.38 | 68,811.54 |

**Table 7.** Experiment result comparison GA-PLAN/OPT.

| Parameter | Time Calculation | Distance (m) | Costs (EUR) | Time (min) | Distance (%) | Costs (%) | Time (%) |
|---|---|---|---|---|---|---|---|
| | 1 min | −98,565.44 | −311.50 | −438.72 | −14.71 | −1.21 | −0.63 |
| Comparison | 5 min | −123,112.02 | −645.48 | −2598.60 | −19.43 | −2.51 | −3.70 |
| GA−PLAN/OPT | 10 min | −107,246.44 | −458.48 | −1439.97 | −19.54 | −1.81 | −2.09 |
| | 20 min | −99,429.15 | −458.52 | −1596.66 | −18.77 | −1.82 | −2.32 |

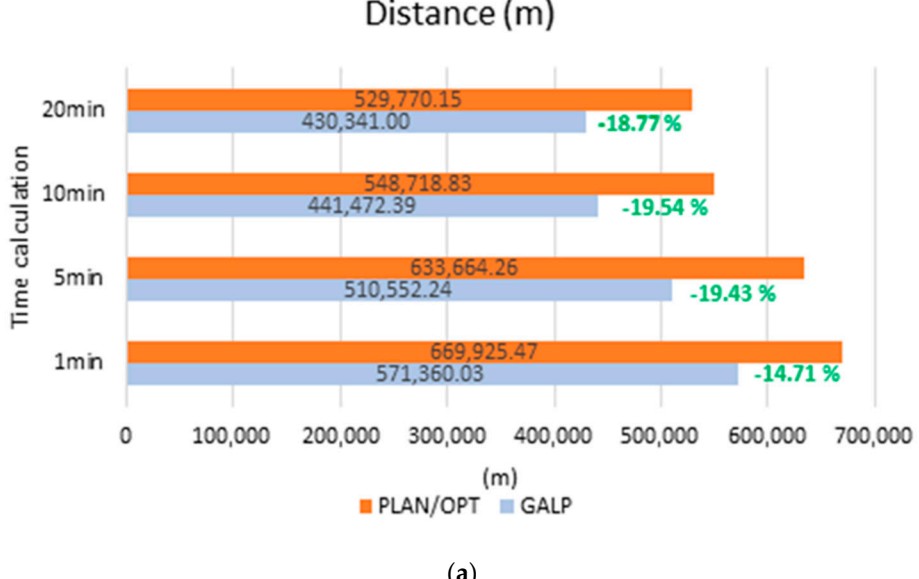

(**a**)

**Figure 11.** *Cont.*

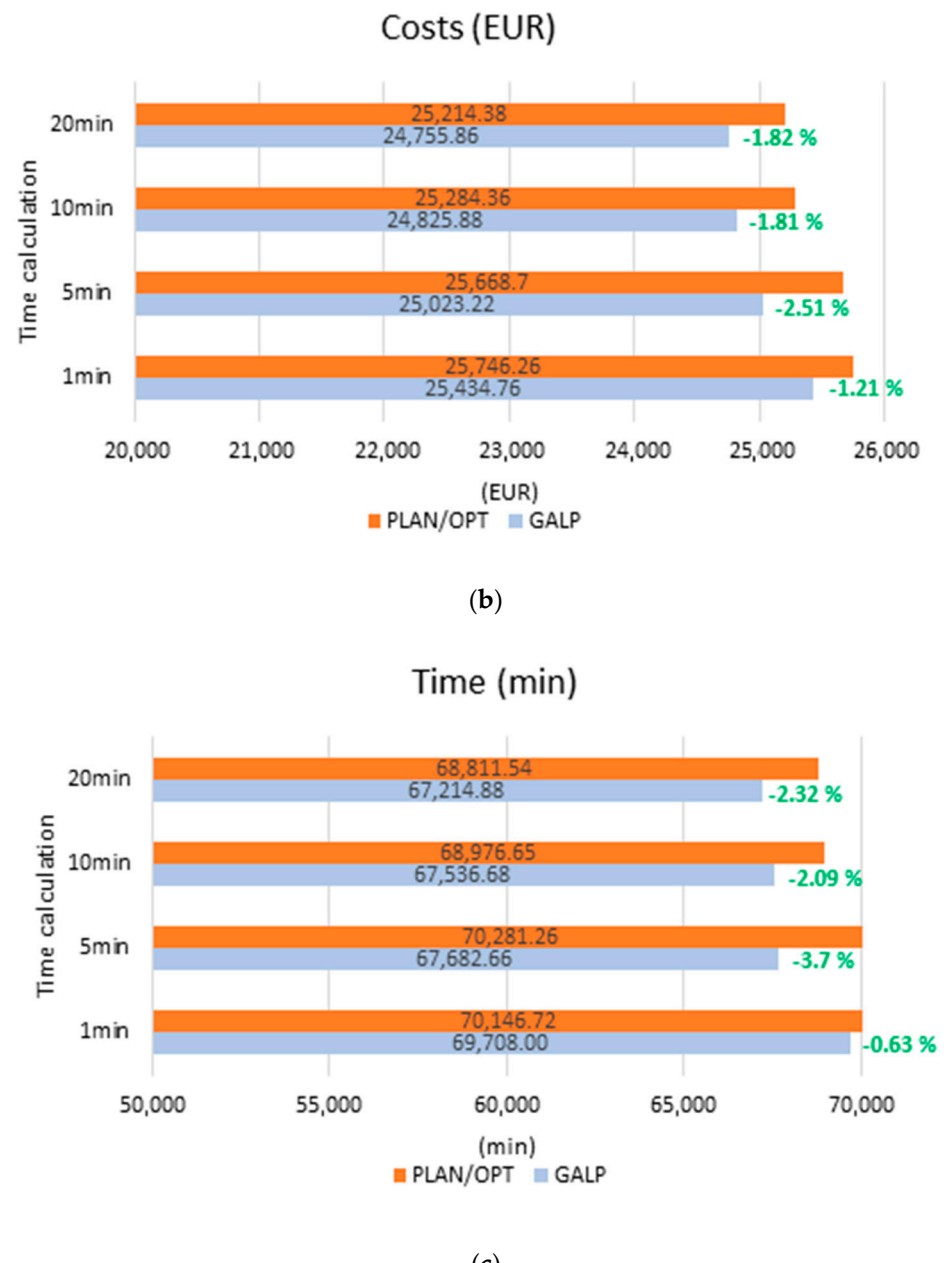

(**b**)

(**c**)

**Figure 11.** Expression of experiment results PLAN/OPT and GALP: (**a**) distance; (**b**) costs; (**c**) time.

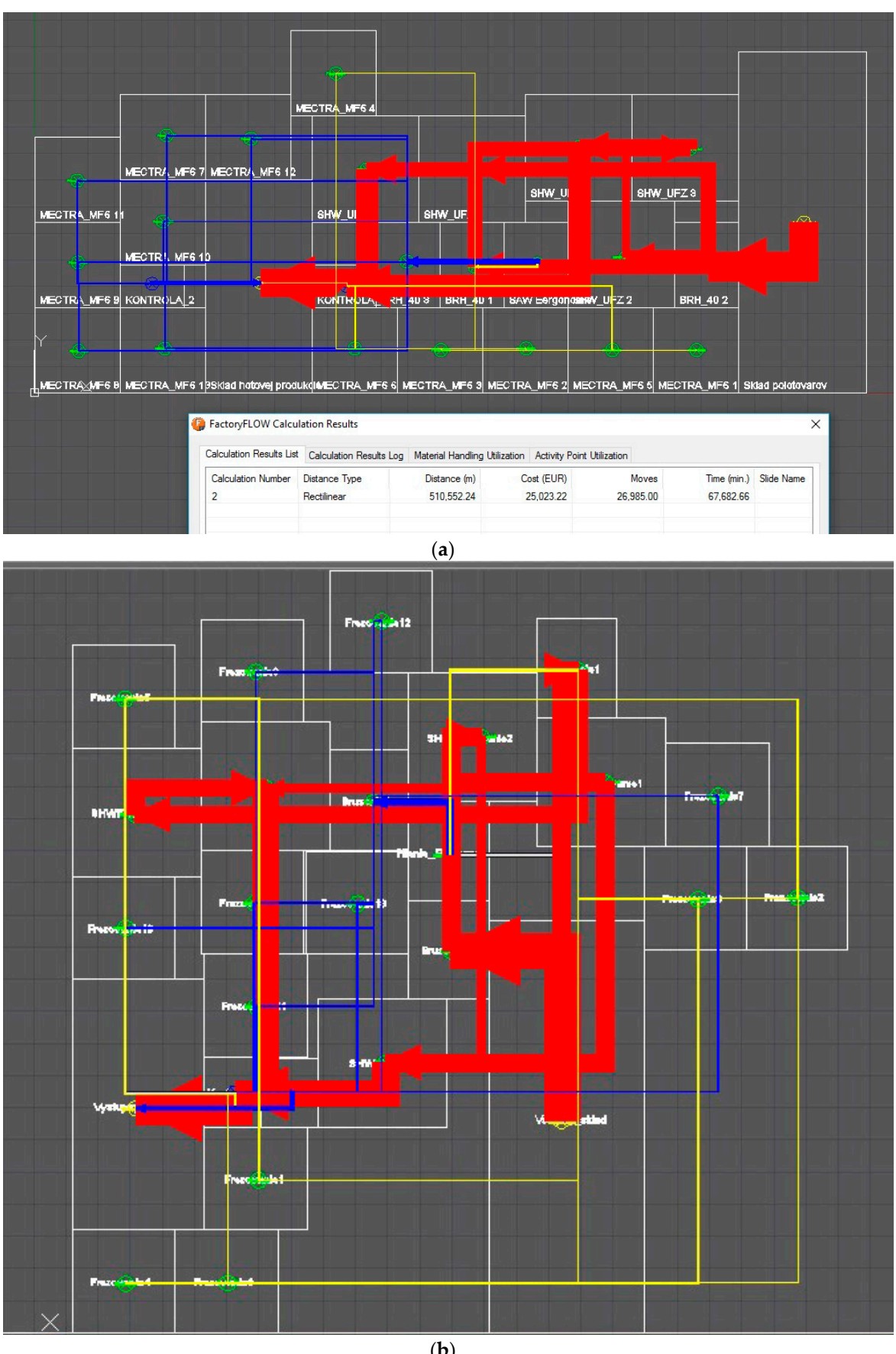

**Figure 12.** *Cont.*

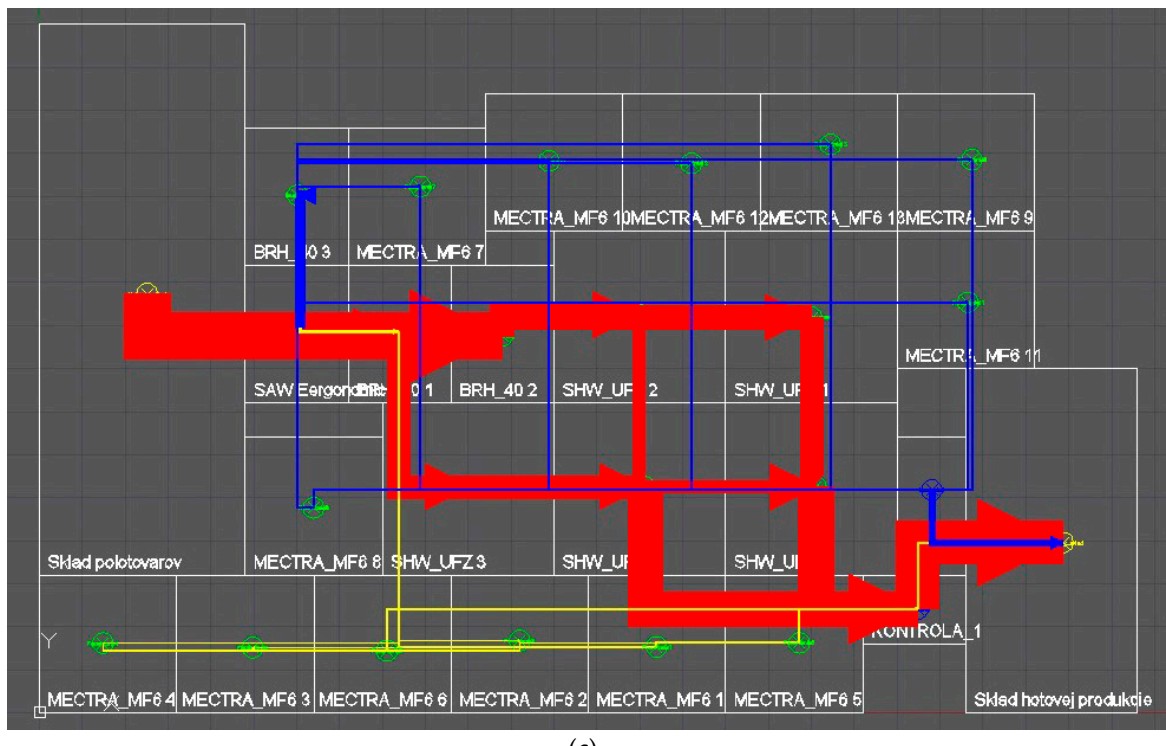

(**c**)

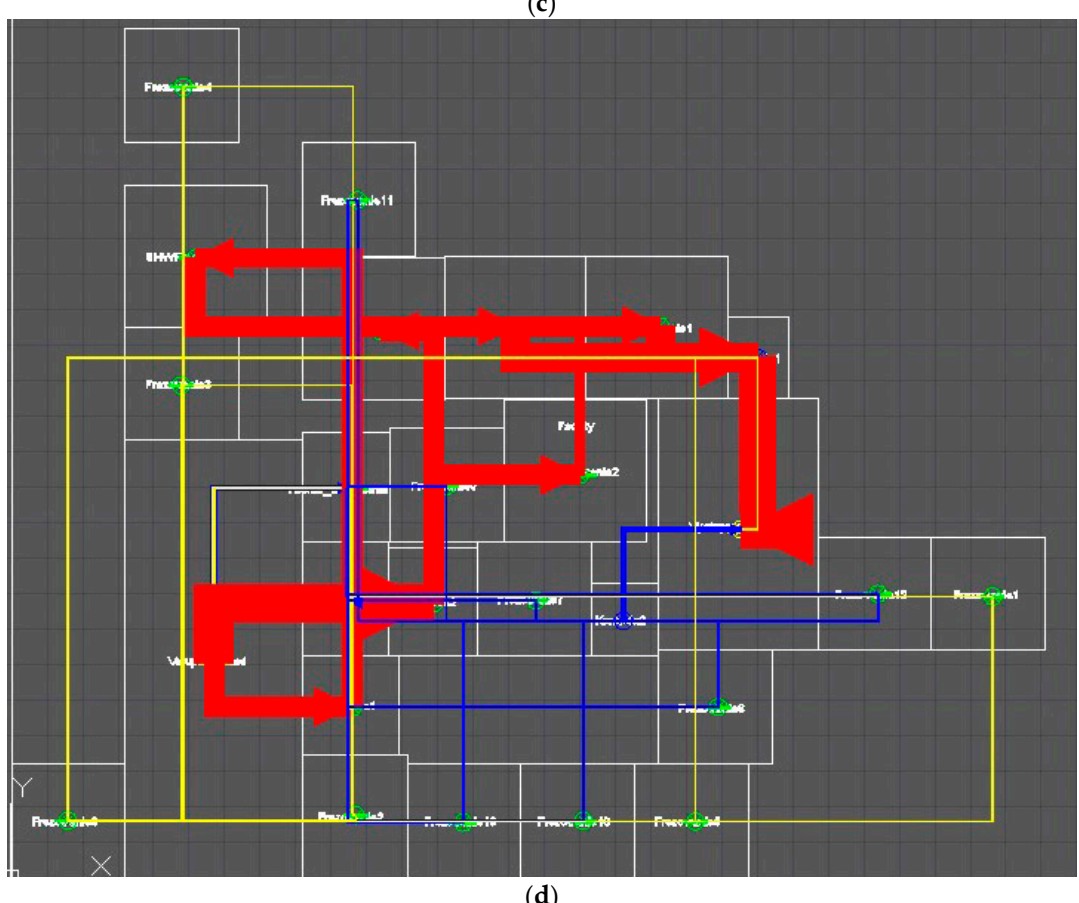

(**d**)

**Figure 12.** Comparison for GALP and PLAN/OPT heuristics: (**a**) GALP (time limit for optimization = 5 m);
(**b**) PLAN/OPT (time limit for optimization = 5 min); (**c**) GALP (time limit for optimization = 20 min);
(**d**) PLAN/OPT (time limit for optimization = 20 min).

The graphical expression of comparison results (Figure 11) shows that the proposed GALP algorithm achieved better results than classical heuristics PLAN/OPT. The results support the decision of the choice of a genetic algorithm as a tool for disposition layout planning and, thus, its applicability in industry.

Experiment results were also consequently verified taking into account the complex solution of a manufacturing system (Case 2) and solution time was set to 5.5 h (solution time 1000 task generations in GALP). Advantages of the genetic algorithm became evident in a more extensive problem. Final material flow is directed with a minimum crossing. However, in case of the PLAN/OPT algorithm, there is a crossing, where material flow keeps coming back, and there is not any technology island creation in the manufacturing system. The genetic algorithm proposed layout with a greatly lower value of transportation performance (38.18%) than the heuristic algorithm in a PLAN/OPT module. Experiment result comparison is stated in Table 8. Final block layouts are shown in Figure 13 below.

**Table 8.** Experiment result comparison for Case 2.

| Parameter | GALP | PLAN/OPT | Difference | Difference (%) |
|---|---|---|---|---|
| Distance covered (m) | 2,877,483.27 | 4,654,622.41 | −1,777,139.14 | 38.18% |
| Costs (EUR) | 83,939.38 | 91,344.13 | −7404.75 | 8.11% |
| Time (min) | 230,818.14 | 253,032.38 | −22,214.24 | 8.78% |

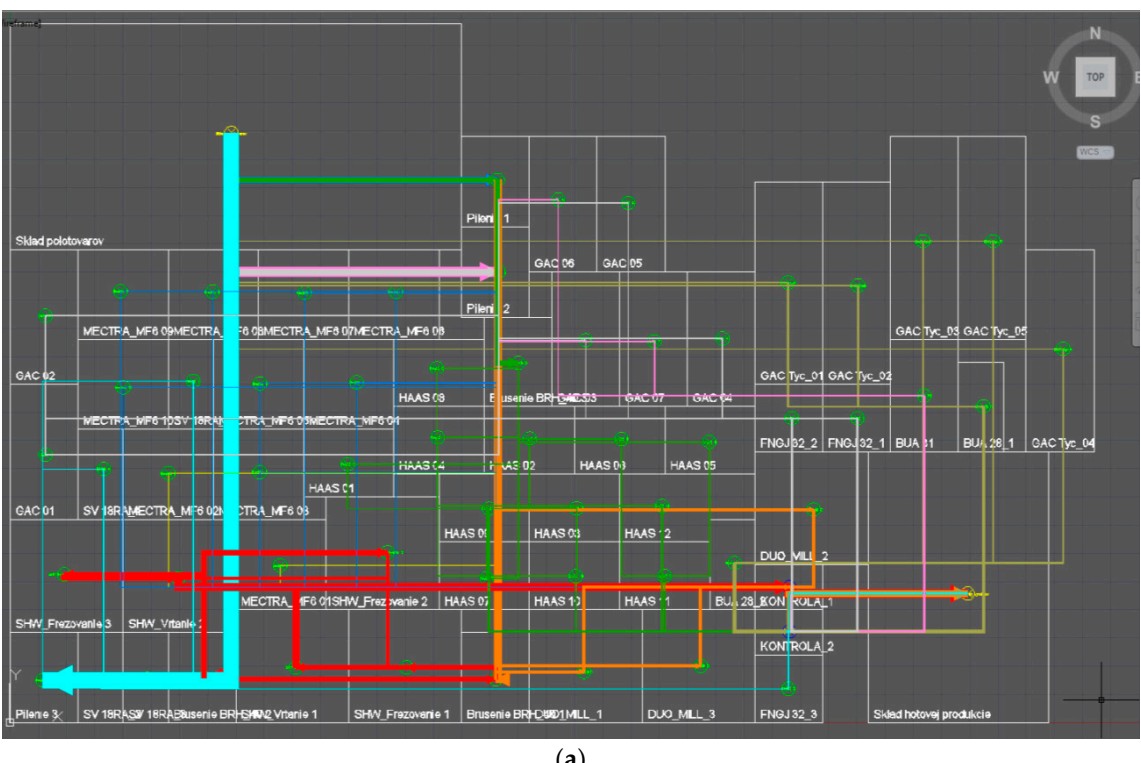

(**a**)

**Figure 13.** *Cont.*

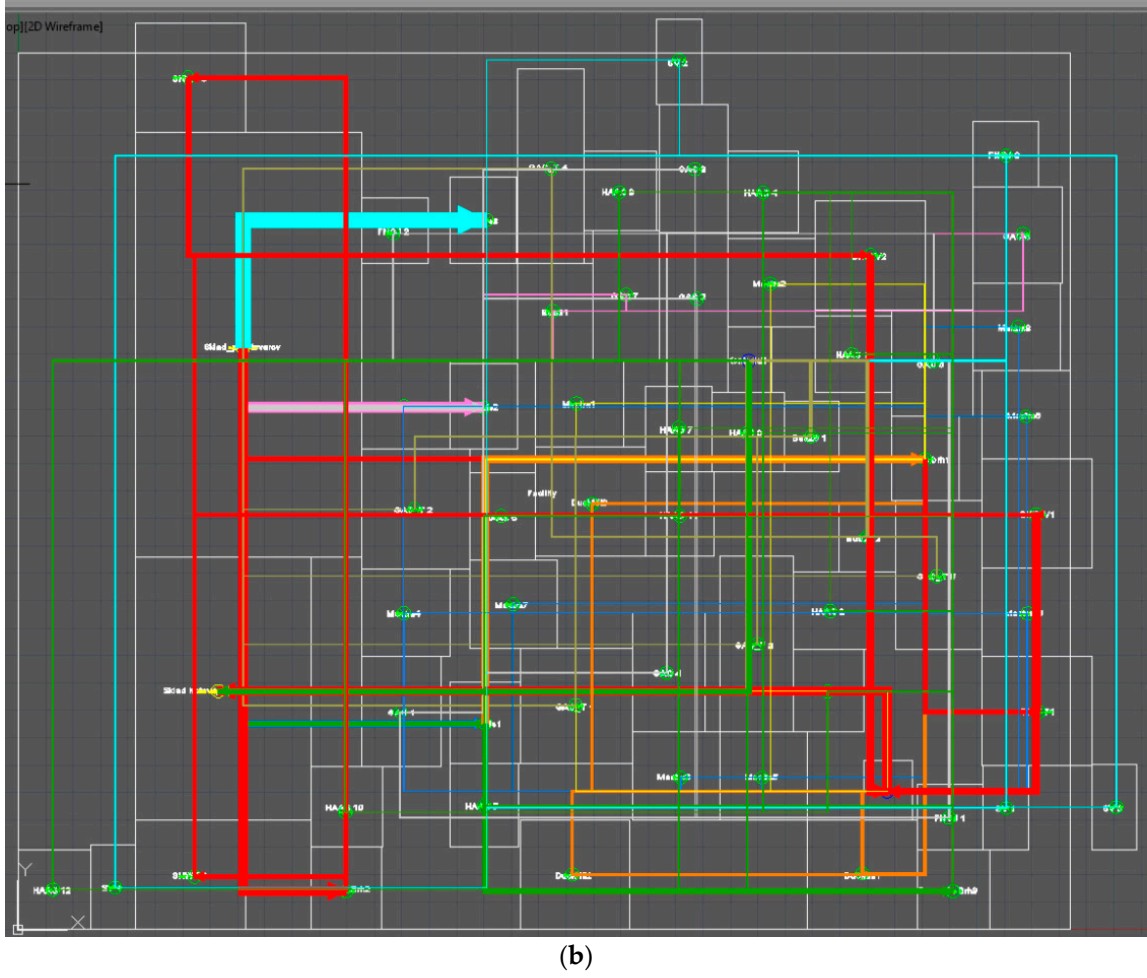

(**b**)

**Figure 13.** Block layout comparison for Case 2: (**a**) Layout generated by the algorithm GALP; (**b**) Layout generated by the PLAN/OPT algorithm.

## 4. Discussion

The main findings of this study are:

- **Application of genetic algorithms for layout design provides better results than traditional heuristics:** As part of the experimental verification, results of the GALP algorithm application were compared with the results of the standard heuristics used for solving layout optimization problems. Heuristics were selected according to Murat (Sequence–Pair approach) because this method is implemented in the software module PLAN/OPT, which is part of the Tecnomatix FactoryFLOW software. Using the above method has enabled the automation of the process of layout variants generation with using traditional heuristics, and software background PLAN/OPT has provided comparable output types as the proposed GALP algorithm. As results of experimental verification of the proposed algorithm show, the GALP application leads to a better layout of the production system. It can be seen on the length of the material flows, on total transport performance, costs and even from point of view of material flow transparency in the resultant layout. In all the GALP experiments led to solutions which resulted in a simple and guided material flow and a lower value of total transport costs. The savings in transport performance and transport costs increase directly in proportion to the growing complexity of the proposed layout.
- **The designed fitness function must be sufficiently comprehensive to ensure that the resulting layout respects the essential requirements for the correct layout:** The fitness function within the

proposed GALP algorithm was designed as a complex function taking into account, on the one hand, the material flow intensity; on the other side relationships among objects (workstations). Such a complex design of fitness functions has allowed the inclusion of other factors in the solution, in addition to the material flow, which affect the design of an optimal spatial structure, such as multi-machine service, teamwork, job rotation, sharing of common documentation, requirements for accuracy and quality of production, requirements for safety and hygiene of work, etc.

- **The proposed GALP algorithm provides a number of advantages in addressing the practical problems associated with the design of the production layout:** Created applications for creating a production disposition that is based on the GALP algorithm simplifies the process of optimizing layout particularly in the variant design phase. A simple user interface for entering input data and visualizing outputs allows designers to realize a set of experiments with different variants of spatial arrangement. The application provides tools for evaluating and comparing individual variants while allowing the static design to be subsequently linked to dynamic solution verification. The practical benefits of the GALP application can be summarized as follows: reducing the time needed to production layout design; reducing the cost for production layout design; verifying a large set of solutions; taking into account restrictive conditions; improving the quality of the proposed layout.

## 5. Conclusions

The main aim of this article was to describe the use of the genetic algorithm in manufacturing layout optimization for continual and sustainable development of the company. The article described not only the basic algorithm structure but also the experimental setting of optimal algorithm parameter and verification. It also compares algorithm outputs of a traditional heuristic application. Experiment results showed the proposed GA provided a saving of transport performance in the case of less complex problems, which was 15 to 20% compared to classical heuristic results. When the problem complexity increases, savings from the GA use continued to increase (Case 2—saving more than 38%). In addition, the disposal arrangement generated by GA leads to a solution with easier and directed material flow. A proposed genetic algorithm is a part of a complex project methodology of manufacturing dispositions, and its basic steps are described in Chapter 3. Furthermore, this proposed GA enables the user to consider practical restrictions when arranging space in layout optimization, that is a shape and production hall dimensions (length, width, and height) building block placement (e.g., columns), fixed installations, transport corridors, input and output spaces of the manufacturing system, etc. Therefore, this means that a layout has been designed by a genetic algorithm requiring the minimum number of corrections that do not represent significant deviations from optimal parameters of material flows. A layout which is designed by the genetic algorithm is ideal not only for stable development and decrease of cost but also for implementation of those solutions which are sustainable in a long time. That means that when we use an ideal layout, it is much easier to maintain optimization changes making them sustainability in a long time.

**Author Contributions:** All authors contributed to writing the paper. Documented the literature review, analyzed the data and wrote the paper. All authors were involved in the finalization of the submitted manuscript. All authors read and approved the final manuscript.

**Funding:** This work was supported by the Slovak Research and Development Agency under contract No. APVV-16-0488.

**Conflicts of Interest:** The authors declare no conflict of interest.

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
