# Peer review of "Parameter Setting for a Genetic Algorithm Layout Planner as a Toll of Sustainable Manufacturing"

_sustainability, doi:10.3390/su11072083_

Round 1

Reviewer 1 Report

The paper addresses the sustainable manufacturing problem through a genetic algorithm layout planner perspective. I found the abstract to be quite short and does not clearly states the purpose of the paper. Please add more information in the abstract. Also, I believe that Introduction section should be re-written as it presents information related to other papers from the field with no particular connection among them. In the current form, it seems that the introduction is just a collection of information which can be re-arranged in any form and it will not change the outcome. Please present the papers by providing a flow to the reading process. Figure 1 and Figure 3 and 4 should be made in the same way, having the same style and colors. Please use lower cases for the text in line 310. Redraw Figure 6 in order to look more academic, having the same style for all the letters and numbers used. Figure 9 should be enlarged in order to increase its readability. Please explain the colors within the figure in a legend. The same for Figure 10, 11. Last, I think that the paper provides a lot of well-known knowledge in the field and this part should be reduced. There are a lot of sections among the paper where the information is presented using bullets  - this is not a common practice for the research paper. Please try to make you paper sound more academic. Also, more space should be given to the numerical example - please present how have you selected the data, its main characteristics, how have you calibrated it, and why the proposed approach is better or comparable to the ones in the field? You can try to test them on similar data sets and compare them in terms of absolute or relative errors. Thank you!

Author Response

Dear Mr. / Mrs. Reviewer

Thank you very much for the inspiring comments on our article. We hope that the modified version of the article will satisfy all your expectations. In view of the necessity to incorporate the comments of other reviewers into the article, the following changes have been made to the article:

1.       Modified and expanded abstract.

2.       Complemented and modified Introduction.

3.       Edited figures as instructed by reviewers.

4.       Added part of the article about the software support to the GALP algorithm.

5.       Completed description for the experimental verification (Numerical Example).

6.       Completed part of the discussion according to reviewers instructions.

7.       Check and edit the English language.

In view of the adjustments made, the list of literature was also modified.

Kind regards, collective of Author's

Reviewer 2 Report

The paper “Parameter setting for Genetic Algorithm Layout  Planner as a toll of sustainable manufacturing” analysis the impact of parameters on the performance of GALP. The sustainability aspects of the article are not clear. Nonetheless I have to confess that I do not see, for the time being, a clear link between this paper's very identity and the journal's overall philosophy.

Title: The title of the paper is informative. It includes important terms and the message of the article.

Abstract: The abstract describes the context and follow the structure: backgrounds, methods, results and conclusions.

Keywords: OK.

Introduction. Introduction defines the focus and the research questions.

Literature review: The literature review supports to understand the correlation of presented research results with literature.

Materials and methods. The methodology is extensively discussed. The case study validates the model. The key functionality has been explained but the computer application for implementation is not revealed. The details of software engineering part would be interesting for anyone aiming to replicate the implementation. Pseudo-code would be useful, if suitable.

Discussion and conclusions: This is the critical part of the article.

·         Application of genetic algorithms for layout design provides better results than traditional heuristics…”: You should compare the GALP with other heuristics suitable for NP-hard layout design problems.

·         “Activity and results of application of genetic algorithm depend on the correct adjustment of the basic algorithm parameters…”. In my opinion, this is not a new findings. This is a fact in the case of all heuristic solutions.

Author Response

Dear Mr. / Mrs. Reviewer

Thank you very much for the inspiring comments on our article. We hope that the modified version of the article will satisfy all your expectations. In view of the necessity to incorporate the comments of other reviewers into the article, the following changes have been made to the article:

1.       Modified and expanded abstract.

2.       Complemented and modified Introduction.

3.       Edited figures as instructed by reviewers.

4.       Added part of the article about the software support to the GALP algorithm.

5.       Completed description for the experimental verification (Numerical Example).

6.       Completed part of the discussion according to reviewers instructions.

For your question:

(Activity and results of application of genetic algorithm depend on the correct adjustment of the basic algorithm parameters: …)

That fact was mentioned in the discussion of the results because the setting of the optimal parameters of the genetic algorithm was part of the research in the area. Optimal parameter values have been set based on experimental examination. The results of the experiments are listed in Table 3 and Table 4.  The resulting values of the genetic algorithm parameters have been selected based on the speed of the convergence of the solution and the best value of a comprehensive fitness function.

7.       Check and edit the English language.

In view of the adjustments made, the list of literature was also modified.

Kind regards, collective of Author's

Reviewer 3 Report

Dear authors, I was pleased to read your study and I believe it raises interesting insights. It provides a detailed plan for the subsequent adoption of genetic algorithms for layout projecting. However, before I recommend its publication, it has several limitations to be overcome: In this version, the introduction section is very similar to a literature review section. Therefore, author/s should focus more on the need of the study, the novelty of this work and the choice of the methodology. Moreover, it emerges the need to highlight the associated research question in the introduction section. I would suggest to improve the literature section. When deepening you theoretical argumentation you may, in every section, start from a general standpoint before focusing on the specific context. My suggestion is to provide in the introduction a more comprehensive definition of digital factory and layout optimisation problems in the context of digital factory according to the body  of literature to justify the research framework and the need of a similar study. You suggest to consider the most adopted definition of digital factory  provided by Centobelli et al. (2016) and Kuhn (2006) (https://en.wikipedia.org/wiki/Digital_factory). Centobelli et al. (2016). "Layout and Material Flow Optimization in Digital Factory". International Journal of Simulation Modelling. 15 (2): 223–235. doi:10.2507/ijsimm15(2)3.327. Kuhn (2006). "Digital Factory - Simulation Enhancing the Product and Production Engineering Process". As minor points please improve the quality of figures 3, 6. With regard to the main results, please stress managerial implications of the study. Future research opportunities should also be better emphasized. Good luck for your research.

Author Response

(The authors gave the same response as above.)

Round 2

Reviewer 1 Report

Thank you for the revised version. I have no further comments. 

Author Response

Dear Mr / Mrs. Reviewer

Thank you very much. We are glad that the incorporated suggestions were in line with your ideas. At the same time, thank you for your suggestions on our article, because they helped us to increase the scientific level of the article.

Based on another review, the article was completed with a graphical expression of the results to support the data in the tables.

Kind regards

Author’s collective

Reviewer 2 Report

Authors have made some changes in this article, but in  my opinion, the scientific soundness of the article should be increased focusing on the following topics:

why have you choosen genetic algorithm? There are great literature sources describing layout planning with other heuristics.

Impact of parameter settings on the performance of the algorithm (however it is also described in many sources)

Add something new to increase the scientific soundness of the article to increase the applicability area or performance of the algorithm.

Performance analysis with benchmarking functions (comparison with other heuristics, not only with with other existing layot planning softwares), prove that your findings makes the performance better

Author Response

Dear Mr / Mrs. Reviewer

Thank you very much. We are glad that the incorporated suggestions were in line with your ideas. At the same time, we would like to thank you for further suggestions on our article, as they will help us to increase the scientific level of the article.

Based on your recommendations, the article has been completed with a graphical expression of the results to support the data in the tables. At the same time, a paragraph in the introduction is added to support the decision to deal with the genetic algorithm as a tool for planning production disposition in industrial practice.

Kind regards

Author’s collective
